

# Evolution analysis of *FRIZZY PANICLE* (*FZP*) orthologs explored the mutations in DNA coding sequences in the grass family (Poaceae)

Jia Li[1,*], Litian Zhang[2,3,*], Rania G. Elbaiomy[4], Lilan Chen[1], Zhenrong Wang[1], Jie Jiao[1], Jiliang Zhu[5], Wanhai Zhou[1], Bo Chen[1], Salma A. Soaud[6], Manzar Abbas[1], Na Lin[1] and Ahmed H. El-Sappah[1,6]

[1] Faculty of Agriculture, Forestry and Food Engineering, Yibin University, Yibin, Sichuan, China
[2] Academy of Animal Science and Veterinary Medicine, Qinghai University, Xining, Qinghai, China
[3] State Key Laboratory of Plateau Ecology and Agriculture, Xining, Qinghai, China
[4] Faculty of Pharmacy, Ahram Canadian University, 6 October, Egypt
[5] Agriculture and Rural Bureau of Zhongjiang County, Deyang, Sichuan, China
[6] Genetics Department, Faculty of Agriculture, Zagazig University, Zagazig, Egypt
[*] These authors contributed equally to this work.

Corresponding author
Ahmed H. El-Sappah,
ahmed_elsappah2006@yahoo.com

## ABSTRACT

*FRIZZY PANICLE* (*FZP*), an essential gene that controls spikelet differentiation and development in the grass family (Poaceae), prevents the formation of axillary bud meristems and is closely associated with crop yields. It is unclear whether the *FZP* gene or its orthologs were selected during the evolutionary process of grass species, which possess diverse spike morphologies. In the present study, we adopted bioinformatics methods for the evolutionary analysis of *FZP* orthologs in species of the grass family. Thirty-five orthologs with protein sequences identical to that of the *FZP* gene were identified from 29 grass species. Analysis of conserved domains revealed that the AP2/ERF domains were highly conserved with almost no amino acid mutations. However, species of the tribe Triticeae, genus *Oryza*, and C4 plants exhibited more significant amino acid mutations in the acidic C-terminus region. Results of the phylogenetic analysis showed that the 29 grass species could be classified into three groups, namely, Triticeae, *Oryza*, and C4 plants. Within the Triticeae group, the *FZP* genes originating from the same genome were classified into the same sub-group. When selection pressure analysis was performed, significant positive selection sites were detected in species of the Triticeae and *Oryza* groups. Our results show that the *FZP* gene was selected during the grass family's evolutionary process, and functional divergence may have already occurred among the various species. Therefore, researchers investigating the *FZP* gene's functions should take note of the possible presence of various roles in other grass species.

## INTRODUCTION

The grass family (Poaceae) consists of many plant species that are essential to mankind, including cereal crops (the main source of grain foods), forage grasses (a valuable source of livestock feed), sugarcane (a key industrial raw material), bamboos, and turf grasses. Grasses produce tillers as well as florets on spikelets, and variation in the number and arrangement of both tillers and spikelets contributes to the great diversity of grass inflorescence structures. In cereal crops, the grains harvested from spikes serve as the main source of grain foods for humans. During the gradual domestication of wild cereal species to cultivars, dramatic changes have occurred in spike morphology (*Lee et al., 2020*). However, it is unclear whether genes that control the characteristics of spike morphology were selected during the evolutionary process.

*FRIZZY PANICLE* (*FZP*) is a key gene that controls spikelet differentiation in rice (*Oryza sativa*). The *FZP* gene is closely associated with rice yields as it prevents the formation of axillary bud meristems and permits the formation of floral meristems in rice spikelets. It encodes an AP2/ERF transcription factor and is the rice ortholog of the maize *branched silkless1* (*BD1*) gene (*Bai et al., 2017*; *Chuck, 2002*; *Komatsu et al., 2003*). *Komatsu et al. (2003)* analyzed the sequences of different *FZP* mutants and found that amino acid mutations produced severe phenotypes in the conserved AP2/ERF domain and the acidic C-terminus region. This indicates that both the AP2/ERF domain and conserved C-terminus region are necessary for proper *FZP* function. Maize *bd1* mutants are characterized by indeterminate branches in place of female spikelets and a series of lateral spikelets in tassels. Besides, *bd1* mutants display similar tassel phenotypes but differ in severity in the ear. Specifically, weaker phenotypic variations initiate fewer branches in the ear and produce a few fertile florets, whereas the most severe phenotypic variations result in the conversion of most initial spikelets to indeterminate branches. In *Brachypodium distachyon*, the inflorescence is an unbranched spike with a limited number of lateral spikelets and terminal spikelets. The *more spikelets1* (*mos1*) mutant of *B. distachyon* has more axillary meristems that can develop into branches with higher-order spikelets produced from inflorescence meristems. It is believed that *MOS1* is an ortholog of *bd1* and *FZP* (*Derbyshire & Byrne, 2013*). The inflorescences or spikes of barley (*Hordeum vulgare*) and common wheat (*Triticum aestivum*) are unbranched (*Bonnett, 1935*; *Bonnett, 1936*) and usually produce one spikelet per rachis node (*Dobrovolskaya et al., 2015*). The inflorescence meristems of barley and wheat also differ in the formation of spikelets (*Derbyshire & Byrne, 2013*). *Dobrovolskaya et al. (2015)* analyzed the structures and functions of three wheat *FZP* orthologs (*WFZP*) and found that coding mutations of *WFZP-D* caused the supernumerary spikelets (SS) phenotype. The most severe phenotypic effect was produced when *WFZP-D* mutations were combined with a frameshift mutation in *WFZP-A*.

The results described above demonstrate that *FZP* and its orthologs play a crucial role in the spike development process of grasses. Mutations in proteins encoded by the gene cause variations in spike shape, which indicates the highly conservative nature of the gene. Grasses possess extremely diverse spike morphologies, and the high grain yield of modern cereal cultivars is the result of long-term domestication of wild grass species. However, it

is unclear whether *FZP* orthologs were selected during the evolutionary process of grasses. In the present study, we identified the *FZP* gene's orthologs in selected grass species and performed evolutionary analysis using bioinformatics methods to determine the mutations in coding DNA sequences (CDSs) during the evolution process of grasses. Results of this study may serve as a theoretical basis for understanding the functional divergence of the gene among different grass species.

## MATERIALS AND METHODS

### Identification of the *FZP* gene in Poaceae

Using the protein sequence of the *FZP* gene (Os07g0669500) from rice as the seed sequence, blastp were performed in the PlantGDB (http://www.plantgdb.org/), Phytozome v12.1 (https://phytozome.jgi.doe.gov), and NCBI (http://www.ncbi.nlm.nih.gov/) databases for orthologs contained in Poaceae species (the corresponding CDSs were also saved).

### Analysis of conserved motifs and multiple sequence alignment of the *FZP* gene in Poaceae species

Using the online MAFFT (Multiple Alignment using Fast Fourier Transform) multiple sequence alignment program (https://www.ebi.ac.uk/Tools/msa/mafft/), multiple sequence alignment was performed for the protein sequences and CDSs of the Poaceae species. Alignment results obtained from MAFFT were corrected using MACSE (for CDS) (*Ranwez et al., 2011*), a built-in alignment software equipped with the PhyloSuite v1.2.1 software (*Zhang et al., 2020*), and used in subsequent analysis. Use Jalview v2.11.1.4 software (*Waterhouse et al., 2009*) to beautify the multiple sequence alignment files.

### Phylogenetic analysis of the *FZP* gene in Poaceae species

The Models function in the MEGA X (Molecular Evolutionary Genetics Analysis X) software (*Kumar et al., 2018*) was used to search for the optimum model for the generation of a maximum likelihood (ML) tree, used full-length CDS alignment sequence of *FZP* orthologs gene (*El-Sappah et al., 2021a*). Subsequently, the Phylogeny function was used for the construction of the ML tree and improvement of its visual appearance.

Use Gblocks 0.91b (*Talavera & Castresana, 2007*) to select the sequence conserved region of the above full-length CDS alignment sequence, MrModeltest 2.3 (*Nylander, 2004*) selects the optimized nucleotide substitution model and calculates the relevant parameters, Mrbayes 3.04b (*Ronquist et al., 2012*) reconstructs Bayesian tree of *FZP* genes. In the process of rebuilding the Bayesian tree, 4 Markov chains are set up, starting with a random tree, running a total of 2 000 000 generations, sampling once every 100 generations and disposing of 25% of the aging samples, and constructing them based on the remaining samples consensus tree, and compute the posterior probability.

### Selection pressure analysis of the *FZP* gene in Poaceae species

We used codeml (*Yang, 2007*), as implemented in a preset model of EasyCodeML v1.2 software (*Gao et al., 2019*), to test for natural selection in each of the three sub-groups with the Branch-site model using the CDS alignment results and constructed ML tree from the processes described in the previous two sections. This model allows the ratio

of nonsynonymous to synonymous substitution rates ($\omega$) to vary among sites (between sites and branches) to test for site-specific evolution in the context of phylogeny. For each gene region, we evaluated which of codon substitution models (Model A vs. Model A null, Model A: fix_omega = 0, omega = 2; Model A null: fix_omega = 1, omega = 1) was favored using likelihood ratio tests with posterior probability ($p > 0.95$). In addition, we use the custom model of EasyCodeML v1.2 software to set the model parameters and use different $\omega$ values to test the robustness of the analyses result.

### Predict the impact of amino acid substitutions or indels on the biological functions of *FZP* gene protein

PROVEAN (Protein Variation Effect Analyzer) is a software tool that predicts whether an amino acid substitution or indel impacts the biological function of the protein. PROVEAN is useful for filtering sequence variants to identify nonsynonymous or indel variants that are predicted to be functionally important. We utilize the PROVEAN online server (http://provean.jcvi.org/seq_submit.php), which predicts whether an amino acid indel or substitution of *FZP* gene (Os07g0669500) affects the biological function. Software operation reference NCBI nr (September 2012) database.

### Protein 3D structure prediction

We use the I-TASSER online server (https://zhanggroup.org/I-TASSER/) to predict the 3D structure of the target gene. Then we did use PyMOL v2.5.0 software (*Schrodinger, 2015*) to visualize according to the structure file of the prediction result (*El-Sappah et al., 2021b*).

## RESULTS AND ANALYSIS

### Identification of the *FZP* gene in Poaceae species

Thirty-five *FZP* orthologs were identified from the sequenced genomes of 29 Poaceae species (Table 1) (Data S1 and S2). *Zea mays* (maize) possessed two *FZP* orthologs (GRMZM2G458437 and GRMZM2G307119) located 44,369,044 bp apart on chromosome 2. *T. aestivum* possessed three *FZP* orthologs (TraesCS2A02G116900.1, TraesCS2B02G136100.1, and TraesCS2D02G118200.1) separately located on chromosomes 2A, 2B, and 2D. *Triticum dicoccoides* possessed two orthologs (TRIDC2AG014040.1 and TRIDC2BG016990.1) separately located on chromosomes 2A and 2B. *Saccharum spontaneum* possessed two *FZP* orthologs (Sspon.02G0001330-3D and Sspon.02G0001330-2C) separately located on chromosomes 2C and 2D. Species of the genus *Oryza* only possessed a single *FZP* gene sequence on chromosome 7. The model dicot species *Arabidopsis thaliana* did not possess genes with high homology to the *FZP* gene, and the same result was obtained with other dicots.

### Analysis of conserved domains of the *FZP* gene in Poaceae species

A highly conserved AP2/ERF domain is present in all 35 *FZP* orthologs (Fig. 1A and Fig. S1). The domain comprised 59 amino acids, with mutations only occurring at positions 13 (QBI22216.1) and 39 (QBI22219.1) (Fig. 1A), which belong to *Triticum monococcum* and *Triticum turgidum*, respectively. This indicates that the domain was a core domain that had remained highly conserved throughout the evolutionary process. The amino acids of

**Table 1** *FZP* orthologs in 29 Poaceae species.

| Species | Gene ID | Chr | Protein length (aa) | E-value[a] | Identity (%)[b] | Databases |
|---|---|---|---|---|---|---|
| *Aegilops tauschii* | XP_020192385.1 | 2D | 314 | 6e−109 | 71.6 | NCBI |
| *Brachypodium distachyon* | Bradi1g18580.1 (*MOS1*) | Bd1 | 319 | 4e−86 | 57 | Phytozome |
| *Dichanthelium oligosanthes* | OEL14197.1 | – | 305 | 3e−123 | 69.4 | NCBI |
| *Eragrostis curvula* | TVU37303.1 | 4 | 285 | 1e−114 | 70.3 | NCBI |
| *Hordeum vulgare* | KAE8778899.1 | 2H | 290 | 2e−113 | 71.6 | NCBI |
| *Leersia hexandra* | LPERR07G22580.1 | 7 | 328 | 1e−117 | 66 | EnsemblPlants |
| *Oryza barthii* | OBART07G25780.1 | 7 | 213 | 7e−87 | 99 | EnsemblPlants |
| *Oryza Sativa* L. spp. japonica | Os07g0669500 (*FZP*) | 7 | 319 | 0 | 100 | EnsemblPlants |
| *Oryza Sativa* L. spp. indica | EAZ05082.1 | 7 | 299 | 0 | 100 | NCBI |
| *Oryza glaberrima* | ORGLA07G0201100.1 | 7 | 321 | 0 | 98 | EnsemblPlants |
| *Oryza glumaepatula* | OGLUM07G25800.1 | 7 | 320 | 0 | 99 | EnsemblPlants |
| *Oryza rufipogon* | ORUFI07G26720.1 | 7 | 319 | 0 | 100 | EnsemblPlants |
| *Oryza longistaminata* | AAX83534.1 | 7 | 279 | 0 | 99.7 | NCBI |
| *Oryza meridionalis* | OMERI07G22420.1 | 7 | 319 | 0 | 98 | EnsemblPlants |
| *Oryza meyeriana* | KAF0907722.1 | 7 | 292 | 4e−147 | 82.4 | NCBI |
| *Oryza nivara* | ONIVA07G25390.1 | 7 | 319 | 0 | 100 | EnsemblPlants |
| *Oryza punctata* | OPUNC07G24180.1 | 7 | 321 | 6e−163 | 90 | EnsemblPlants |
| *Panicum miliaceum* | RLN33328.1 | 3 | 304 | 2e−124 | 70.5 | NCBI |
| *Panicum virgatum* | Pavir.Ba00192.1 | 02a | 302 | 4e−126 | 72 | Phytozome |
| *Panicum hallii* | Pahal.2G484900.1 | 2 | 310 | 2e−128 | 69 | Phytozome |
| *Saccharum spontaneum* | Sspon.02G0001330-3D | 2D | 377 | 1e−115 | 74 | EnsemblPlants |
|  | Sspon.02G0001330-2C | 2C | 378 | 3e−116 | 74 | EnsemblPlants |
| *Setaria italica* | Seita.2G427300.1 | scaffold_2 | 294 | 3e−125 | 71 | Phytozome |
| *Setaria viridis* | Sevir.2G437800.1 | 2 | 294 | 3e−125 | 71 | Phytozome |
| *Sorghum bicolor* | Sobic.002G411000.1 | 2 | 327 | 1e−129 | 71.6 | Phytozome |
| *Triticum aestivum* | TraesCS2A02G116900.1(*WFZP-A*) | 2A | 300 | 2e−109 | 69 | EnsemblPlants |
|  | TraesCS2B02G136100.1(*WFZP-B*) | 2B | 308 | 3e−113 | 71 | EnsemblPlants |
|  | TraesCS2D02G118200.1(*WFZP-D*) | 2D | 296 | 2e−109 | 72 | EnsemblPlants |
| *Triticum dicoccoides* | TRIDC2AG014040.1 | 2A | 300 | 2e−109 | 69 | EnsemblPlants |
|  | TRIDC2BG016990.1 | 2B | 305 | 6e−115 | 68 | EnsemblPlants |
| *Triticum monococcum* | QBI22216.1 | 2A | 299 | 9e−108 | 68.4 | NCBI |
| *Triticum turgidum* | VAH42493.1 | 2B | 307 | 2e−113 | 71.5 | NCBI |
|  | QBI22219.1 | 2A | 299 | 9e−108 | 68.4 | NCBI |
| *Zea mays* | GRMZM2G458437 (*BD1B*) | 2 | 296 | 1e−104 | 62.5 | MaizeGDP |
|  | GRMZM2G307119 (*BD1*) | 2 | 315 | 6e−108 | 66.5 | MaizeGDP |

**Notes.**

[a],[b]Use the rice *FZP* gene (Os07g0669500) protein sequence as the query sequence, and other species protein sequences as the subject sequence using the Blastp program of the NCBI (https://blast.ncbi.nlm.nih.gov/Blast.cgi?PROGRAM=blastp&PAGE_TYPE=BlastSearch&LINK_LOC=blasthome) calculate *E*-value and identity (%).

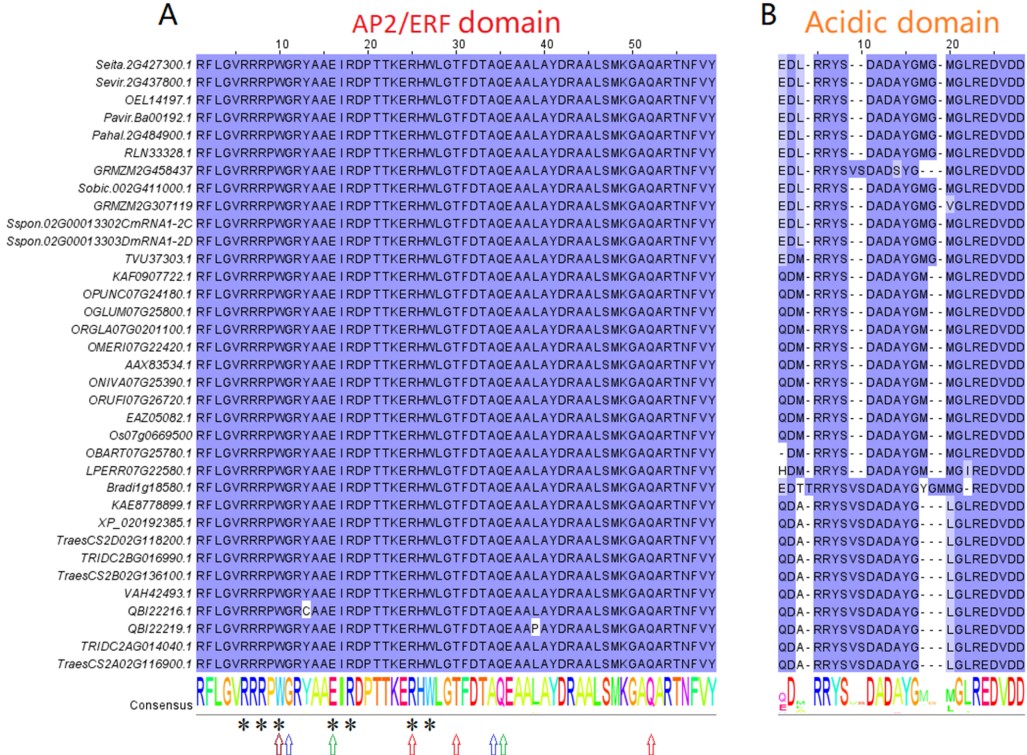

**Figure 1** Conserved amino acid motifs of *FZP* orthologs gene in Poaceae. (A) The AP2/ERF domain of *FZP* orthologs gene. Amino acids residues with asterisks indicate that confer specific GCC box binding. The amino acid residues with brown, blue, green and red arrows indicate mutation sites that can cause supernumerary spikelets (SSs) phenotype in maize, *Brachypodiumdistachyon*, bread wheat and rice, respectively. An amino acid substitution is shown at position 13 (Y13C, Y at position 13 is changed to C) and 39 (L39P, L at position 39 is changed to P) of the QBI22216.1 and QBI22219.1 genes, respectively. (B) The C-terminus acidic domain of *FZP* orthologs gene.

positions 6, 8, 10, 16, 18, 25, 27 (asterisks) indicate that confer specific GCC box binding. In a specific Poaceae species category, the acidic domain of C-terminus was also relatively conserved during evolution. Visible differences exist in the acidic domain of C-terminus of these three categories of Poaceae, which serve as the main domain for distinguishing between them (Fig. 1B and Fig. S1). For instance, position 3 was occupied primarily by Ala (A) in Triticeae, Met (M) in *Oryza*, and Leu (L) in C4 plants; Positions 8 and 9 were inserted by Ser (S) and Val (V) in Triticeae but were deleted in *Oryza* and C4 plants; the amino acid of position 16 was deletion in Triticeae, and Met (M) in *Oryza* and C4 plants. The amino acid residues of positions 23-28 are very conservative (Fig. 1B).

## Phylogenetic analysis of the *FZP* gene in Poaceae species

Using MEGA X, the optimum nucleotide replacement model T92+G was identified to generate the ML tree for the full-length CDS sequence of *FZP* gene in Poaceae species (Fig. 2) (Data S3). Results indicate that the grass species could be classified into groups I, II, and III. The bootstrap values of the three groups were 98%, 100%, and 66%, respectively, which were >95% and indicative of high reliability. Group I consisted of species of the

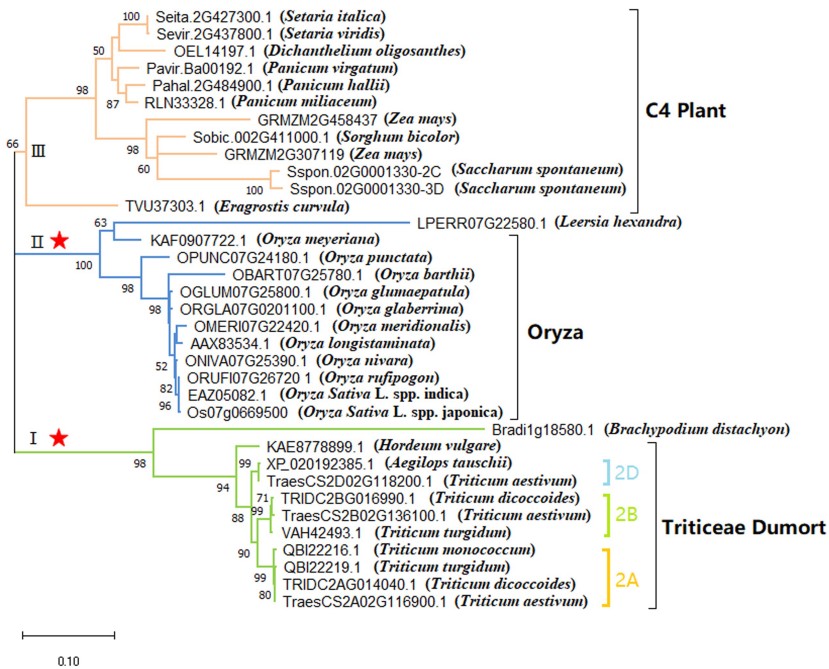

**Figure 2** Phylogenetic analysis of the *FZP* orthologs gene in Poaceae (ML tree). The numbers above the branches indicate the bootstrap values. The branch labels are tagged with gene ID numbers and Latin names of species (inside brackets). The red asterisks indicate the branch with sites of significant positive selection.

tribe Triticeae and genus *Hordeum*, which were further divided into three sub-groups 2A, 2B, and 2D. Subgroup 2A consisted of four *FZP* genes separately located on chromosome 2A of *T. aestivum*, *T. dicoccoides*, *T. turgidum*, and *T. monococcum*. Subgroup 2B consisted of three *FZP* genes separately located on chromosome 2B of *T. aestivum*, *T. dicoccoides*, and *T. turgidum*. Subgroup 2D consisted of two *FZP* genes separately located on chromosome 2D of *Aegilops tauschii* and *T. aestivum*. Group II was comprised entirely of species of the genus *Oryza* and had a bootstrap value of 100%, which indicates the extremely high reliability of the group. Except for the C3 grass *Dichanthelium oligosanthes* (OEL14197.1) (*Studer et al., 2016*), all other species in Group III were C4 plants, which included three species of the genus *Panicum* (*P. virgatum* (*Flint et al., 2019*), *P. hallii* (*Milano et al., 2018*), and *P. miliaceum* (*Son & Sugiyama, 1992*)) and two species of the genus *Setaria* (*S. italica* (*Heimann et al., 2013*) and *S. viridis* (*Brutnell et al., 2010*)). The group also consisted of two genes from *Z. mays* (*BD1* and *BD1B*), one gene from *S. bicolor* (*Paterson et al., 2009*; *Tazoe et al., 2016*), and two genes from *Saccharum spontaneum* (*Zhang et al., 2018*). Previous research has indicated that C3 plants of Poaceae from the subfamilies Ehrhartoideae and Pooideae and C4 plants of Poaceae from the subfamily Panicoideae had diverged over 50 million years ago. Still, the C3 plants of Poaceae *D. oligosanthes* only separated from the C4 lineages represented by *Setaria viridis* and *S. bicolor* approximately 15 million years ago (*Studer et al., 2016*). This shows that the *FZP* orthologs can effectively distinguish between C3 and C4 grasses that diverged earlier, but not between C3 grasses whose divergence

occurred more recently. In other words, mutations had already occurred in the amino acid sequence of the gene during the early divergence of C3 and C4 grasses. Although amino acid mutations in the AP2/ERF domain have been rare, distinct mutations have occurred in the C-terminus acidic domain, as shown in Fig. 1.

After multiple alignments, redundant sequences were cut to obtain a CDS sequence conserved region with a length of 438 bp. According to the AIC standard of MrModeltest software, the optimum nucleotide replacement model for BI tree is F81+G, and the rate of variation between sites rates = gamma (Lset nst = 1 rates = gamma; Prset statefreqpr = Dirichlet (1, 1, 1, 1)), The *FZP* gene Bayesian phylogenetic tree reconstructed by Bayesian inference (BI) tree is shown in Fig. S2 and supplementary data 4. The topology is consistent with the ML tree.

### Detection of selection pressure of the *FZP* gene in Poaceae species

The three branches; Triticeae, *Oryza*, and C4 plants, were set separately as the foreground branch (Fig. 2) and all other branches were set as the background branches for the analysis of selection pressure of the *FZP* gene in grasses using the branch-site model in EasyCodeML v1.2. When Triticeae was marked as the foreground branch, the likelihood ratio test (LRT) was highly significant ($P = 0.000017894$). Two positive selection sites were detected, with a significant positive selection site (61 I, posterior probability >95%) (Table 2 and Fig. S1) located near the upstream region of the AP2/ERF domain. When the *Oryza* branch was marked as the foreground branch, the LRT was significant ($P = 0.030827376$). Three positive selection sites were detected, including a significant positive selection site (484 G, posterior probability >95%) (Table 2) located at the C-terminus. When the C4 plants branch was marked as the foreground branch, the LRT was not significant ($P = 0.127622034$). Six positive selection sites were detected, including 401 E sites within the acidic domain of C-terminus and one significant positive selection site (328 S, posterior probability >99%) (Table 2 and Fig. S1) near the upstream region of the acidic domain of C-terminus.

### Predict the impact of amino acid substitutions or indels on the biological functions of *FZP* gene protein

In order to understand whether an amino acid substitution or indel has an impact on the biological function of protein. We used PROVEAN online server (http://provean.jcvi.org/seq_submit.php) to predict amino acid substitutions or indels of *FZP* gene (Os07g0669500) on the biological functions. The results show that the significant positive selection sites (sites with asterisks) detected in 3.4, PROVEAN scores are above −2.5, variants are considered "neutral" (Table 3). However, the Y69C (score = −8.925) and L95P (score = −6.242) amino acid substitutions in the AP2/ERF domain of QBI22216.1 and QBI22219.1 gene are considered "deleterious" (Table 3). The nine amino acid deletions in the C-terminus of the AAX83534.1 gene in *Oryza longistaminata* (Y310_H318del, score = −4.189) are considered "deleterious" (Table 3).

Li et al. (2022), *PeerJ*, DOI 10.7717/peerj.12880

**Table 2 Selection pressure analysis of the *FZP* orthologous gene in Poaceae using the branch-site model.**

| Branch | Model | np[a] | Ln L[b] | | Estimates of parameters | | | | LRT *P*-value | Positive sites[c] |
|---|---|---|---|---|---|---|---|---|---|---|
| | | | | Site class | 0 | 1 | 2a | 2b | | |
| Triticeae Branch | Model A | 73 | −7889.229233 | f | 0.72446 | 0.23826 | 0.02805 | 0.00923 | 0.000017894** | **61 I 0.978***, 204 A 0.677, |
| | | | | ω0 | 0.07414 | 1.00000 | 0.07414 | 1.00000 | | |
| | | | | ω1 | 0.07414 | 1.00000 | 675.59953 | 675.59953 | | |
| | Model A null | 72 | −7898.429883 | 1 | | | | | | Not Allowed |
| Oryza Branch | Model A | 73 | −7897.344045 | f | 0.72693 | 0.25530 | 0.01315 | 0.00462 | 0.030827376* | 129 A 0.510, **484 G 0.975*** |
| | | | | ω0 | 0.07428 | 1.00000 | 0.07428 | 1.00000 | | |
| | | | | ω1 | 0.07428 | 1.00000 | 999.00000 | 999.00000 | | |
| | Model A null | 72 | −7899.675320 | 1 | | | | | | Not Allowed |
| C4 Plant Branch | Model A | 73 | −7896.076393 | f | 0.71012 | 0.25956 | 0.02221 | 0.00812 | 0.127622034 | **328 S 0.998****, 392 S 0.682, 393 S 0.689, 401 E 0.621, 443 D 0.530, 444 A 0.844 |
| | | | | ω0 | 0.07223 | 1.00000 | 0.07223 | 1.00000 | | |
| | | | | ω1 | 0.07223 | 1.00000 | 26.74694 | 26.74694 | | |
| | Model A null | 72 | −7897.236990 | 1 | | | | | | Not Allowed |

**Notes.**
[a]Number of parameters including branch lengths.
[b]Likelihood score.
[c]Positive selected sites (posterior probability >0.95).

**Table 3** **Predict the impact of amino acid substitutions or indels on the biological functions of *FZP* gene proteins.**

| Variant[a] | Multiple sequence alignment site[b] | PROVEAN score[c] | Prediction (cutoff = −2.5) |
|---|---|---|---|
| A29I[*] | A61 | −0.219 | Neutral |
| S45A | S129 | −0.686 | Neutral |
| A52T | A140 | 0.05 | Neutral |
| **Y69C** | Y157 | −8.925 | **Deleterious** |
| **L95P** | L183 | −6.242 | **Deleterious** |
| T116A | T204 | −0.733 | Neutral |
| S197A | S327 | 0.747 | Neutral |
| S198D[*] | S328 | −0.628 | Neutral |
| S252_D253insVS | −409, −410 | 1.097 | Neutral |
| M259_M260insG | −418 | 0.258 | Neutral |
| M259_M260delinsL | M417–M420 | −0.342 | Neutral |
| S308_S309insA | −483 | 1.028 | Neutral |
| S308_S309insN | −483 | 0.158 | Neutral |
| S309G[*] | S484 | 0.219 | Neutral |
| S309A | S484 | −0.217 | Neutral |
| **Y310_H318del** | Y485-H493 | −4.189 | **Deleterious** |

Notes.

[a]The variant position number is based on the amino acid sequence of the *FZP* gene (Os07g0669500) without gap as a reference. A29I, A at position 29 is changed to I, other similar types have the same meaning; S252_D253insVS, VS is inserted between positions 252 and 253, other similar types have the same meaning; M259_M260delinsL, MM is replaced by L; Y310_H318del, A deletion of nine amino acids, from Y at position 310 to H at position 318.

Variant with an asterisk indicates a significant positive selection site for selection pressure calculation. Bold variant indicates the "Deleterious" mutation site after amino acid substitution or indel.

[b]The variant position number is based on multiple sequence alignment of *FZP* gene and its orthologous genes (Fig. S1).

[c]Default threshold is −2.5, variants with a score equal to or below −2.5 are considered "deleterious", if above −2.5 are considered "neutral".

## Protein 3D structure prediction

We did use I-TASSER online server (https://zhanggroup.org/I-TASSER/), using default parameters, predicted the protein 3D structure of rice *ZFP* gene (Os07g0669500), wheat *WFZP-A* gene (TraesCS2A02G116900.1), maize *BD1* gene (GRMZM2G307119), *T. monococcum* QBI22216.1 gene and *T. turgidum* QBI22219.1 gene to understand whether amino acid substitutions or indels in the conserved amino acid domain of the *FZP* gene affect the 3D structure of the protein. The results (Fig. 3 and Table S1) show that the 3D structure of these five genes is similar to the *Arabidopsis AtERF96* gene 3D structure (Protein Data Bank [PDB] ID: 5WX9) (*Chen et al., 2020*) (Fig. 3A). There are seven amino acid residues (amino acids with labels in the 3D structure at Fig. 3) that specifically bind to the DNA GCC box at AP2/ERF domain (red $\alpha$-helix, $\beta$-sheet and random coil in 3D structure at Figs. 3A–3F). The 3D structure of *Arabidopsis AtERF96* gene does not have an acidic domain at the C-terminus (Fig. 3A), and gramineous plants all have an acidic domain (the orange $\alpha$-helix, $\beta$-sheet and random coil in the 3D structure at Figs. 3B–3F) at C-terminus (green coil/ $\alpha$-helix in 3D structure at Figs. 3B–3F). The C-terminus acidic domain of gramineous and the AP2/ERF domain are close to each other, especially close to
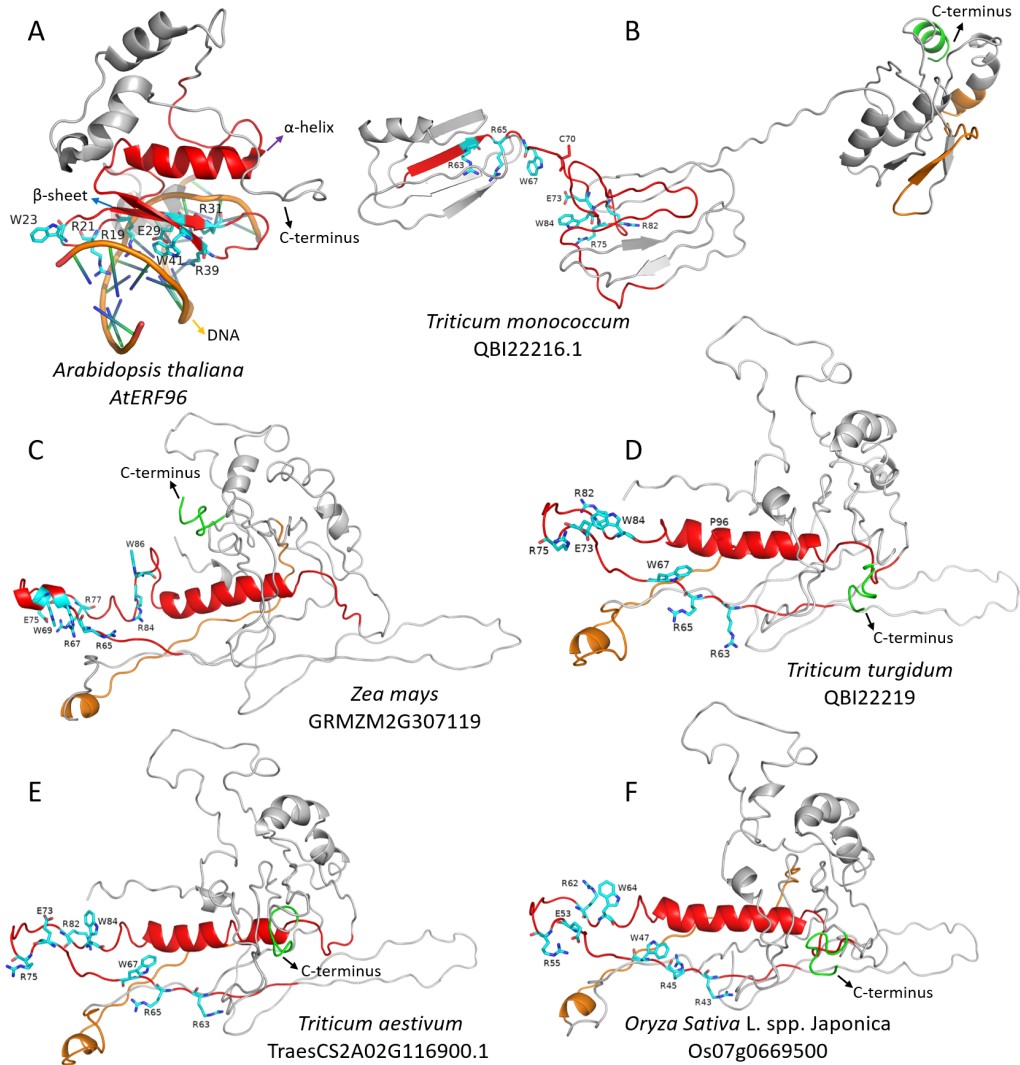

**Figure 3** **The 3D structure of *Arabidopsis AtERF96* gene and *FZP* orthologous gene protein.** The red and orange area in 3D structure indicate N-terminus AP2/ERF domain and C-terminus acidic domain of *FZP* genes, respectively. The C-terminus, α-helix, β-sheet and DNA were point out with black, purple, blue and orange arrows, respectively. The light blue amino acids with labels in the 3D structure indicate that they specifically bind to the DNA GCC box. The red amino acids with labels (P96 and C70) indicate that an amino acid substitution in the AP2/ERF domain are considered "deleterious" (PROVEAN predicted results). (A) Crystal structure of *AtERF96* with GCC-box. (B–F) The 3D structure of *Triticum monococcum* QBI22216.1 gene, *Zea mays BD1* gene (GRMZM2G307119), *Triticum turgidum* QBI22219.1 gene, *Triticum aestivum WFZP-A* gene (TraesCS2A02G116900.1) and *Oryza Sativa* L. spp. japonica *FZP* gene (Os07g0669500). The 3D structure of B–F graphics is visualized using model 1 by PyMOL software.

the region of amino acid residues that specifically bind to the DNA GCC box (except for the QBI22216.1 gene). The substitution of P96 (L183P in Fig. S1) amino acid residue in the AP2/ERF domain of *T. turgidum* QBI22219.1 gene did not cause major changes in the 3D structure of the protein (Fig. 3D). It is worth noting that the substitution of C70 (Y157C in Fig. S1) amino acid residues in the AP2/ERF domain of *T. monococcum* QBI22216.1 gene

causes a huge change in the 3D structure of the protein. The AP2/ERF domain cannot form an $\alpha$-helix, and the C-terminus acidic domain was far away from AP2/ERF domain (Fig. 3B).

## DISCUSSION

In this study, a total of 34 orthologs of the *FZP* gene were identified in 26 grass species. No complete or highly identity *FZP* orthologous genes have been searched in *Triticum urartu*, *Eragrostis tef*, and *Oryza brachyantha*, which are also gramineous plants. There are two possible reasons for these results: (1) The genome assemblies of *T. urartu*, *E. tef*, and *O. brachyantha* may be incomplete; and (2) an additional level of genetic regulation is required for inflorescence formation in grasses (*Komatsu et al., 2003*). Specifically, maize (*Z. mays*) possessed two *FZP* orthologs (*BD1* and *BD1B*), located on chromosome 2. Mutations in the *BD1* gene cause the formation of indeterminate lateral tillers in the ear and tassel, whereas the functions of the *BD1B* gene have not yet been reported. This indicates that the *BD1B* gene maybe a duplicate of *BD1* generated during the maize genome duplication. The common wheat (*T. aestivum*) possessed three *FZP* orthologs (*WFZP*) that separately belonged to the A, B, and D genomes. The results of the evolutionary analysis revealed that the *WFZP-A* gene of common wheat was located on the same branch as the *FZP* ortholog of *T. monococcum*, the *WFZP-B* gene was located on the same branch as the *FZP* ortholog of *T. dicoccoides*, and the *WFZP-D* gene was located on the same branch as the *FZP* ortholog of *A. tauschii*. These results demonstrate that the evolutionary process of the *FZP* gene in common wheat was consistent with the evolutionary theories for common wheat.

Modern theories have generally asserted that common wheat was domesticated in the Fertile Crescent approximately 10,000 years ago (*Lev-Yadun, Gopher & Abbo, 2000*) with its hexaploid genome formed through the hybridization of tetraploid wheat with the wild diploid grass *A. tauschii* (*Abbo et al., 2014*). Except for *T. turgidum*, *T. dicoccoides*, and *Saccharum spontaneum*, which possessed two *FZP* orthologs, all other species had a single *FZP* ortholog that only existed in plants, indicating that the evolutionary process of the genes occurred more recently. Our analysis of the conserved motifs of *FZP* orthologs in the grass species revealed that amino acid mutations were almost absent in the AP2/ERF domain, which indicates the highly conserved nature of the domain during the evolutionary process. It is known that the occurrence of nonsynonymous substitutions in the AP2/ERF domain leads to changes in gene function. However, the rich evolutionary information has been retained through significant mutations in the acidic C-terminus region of the gene in species of the tribe Triticeae, genus *Oryza*, and C4 plants, which can be used for phylogenetic analyses. Interestingly, except for species belonging to the genera *Oryza*, *B. distachyon*, *Eragrostis curvula*, and *Panicum miliaceum*, the *FZP* genes of all other grass species were located on chromosome 2. Previous research has indicated that chromosome 2 of foxtail millet (*Setariaitalica*) was formed by combining chromosomes 7 and 9 of rice. Besides, the researchers observed the occurrence of fusion events in the chromosomes of Sorghum (*Sorghum bicolor*), which led to the deduction that the chromosome fusion events most likely occurred before the divergence of sorghum and foxtail millet. These

chromosomal recombination events form a key basis for species divergence and genetic variation in species (*Bennetzen et al., 2012*; *Zhang et al., 2012*). Besides maize and polyploid grasses, all other grasses only possessed a single *FZP* ortholog. Researchers have also found that chromosome 2 of foxtail millet was formed through the fusion of chromosomes 7 and 9 of rice and that similar fusion events had occurred in the chromosomes of Sorghum, which led to the deduction that the chromosome fusion events most likely occurred before the divergence of sorghum and foxtail millet. Researchers had also discovered a specific chromosome fusion event after the divergence of foxtail millet. These chromosomal recombination events form a key basis for species dissimilarity and genetic variation in species (*Bennetzen et al., 2012*; *Zhang et al., 2012*). During selection pressure analysis of *FZP* orthologs in Poaceae using the branch-site model, significant positive selection sites were distinguished in the Triticeae and *Oryza* branches (Table 2 and Fig. S1), which demonstrates that the *FZP* gene in the tribe Triticeae and genus *Oryza* was decidedly chosen to various extents during the divergence process and may have gone through functional divergence. This may clarify the substantial differences in spike morphology between the two groups of grass species, and specialists should observe such divergence when investigating the functions of the *FZP* gene.

To understand whether positive selection sites and other amino acid substitution or indel has an impact on the biological function of the protein. We use the PROVEAN online server (http://provean.jcvi.org/seq_submit.php), which predicts whether amino acid substitution or indel of *FZP* gene (Os07g0669500) has an impact on the biological function. The results show that the significant positive selection sites (PROVEAN scores are above −2.5) are considered "neutral" (Table 3 and Fig. S1). However, the Y69C (score = −8.925) and L95P (score = −6.242) amino acid substitutions in the AP2/ERF domain of QBI22216.1 and QBI22219.1 gene are considered "deleterious" (Table 3 and Fig. 1A). These results indicate that although positive selection sites have been detected in the Triticeae dumort and Oryza branches, the substitution of these sites may not affect the gene's function. AP2/ERF domain is highly conserved in gene evolution, and the substitution of amino acids may affect gene function. Previous studies have also proved this point. *Komatsu et al. (2003)* studied rice mutants (*fzp-2* and *fzp-7*), they found that one of the 7 amino acids (R25L in Fig. 1A and R25W in Fig. 1A) that specifically bind to the GCC box (*Allen et al., 1998*; *Chen et al., 2020*; *Hao, Ohme-Takagi & Sarai, 1998*) in the AP2/ERF structure is replaced, which can lead to a severe *frizzy panicle* (*fzp*) phenotype. Even if the conserved amino acids that non-specifically bind to the GCC box in the AP2/ERF domain are replaced (T30M in Fig. 1A), it can result in a weaker *fzp* phenotype. It indicates that *FZP* gene may regulate its target gene through GCC box-mediated gene expression. Certain nucleotide variations at the end of the AP2/ERF domain may produce premature stop codons (Q52Stop in Fig. 1A), leading to a severe *fzp* phenotype. *Dobrovolskaya et al. (2015)* detected a mutation in the GCC-box binding site (E16K in Fig. 1A) and a premature stop codon (Q35Stop in Fig. 1A) in the AP2/ERF functional domain of *FZP-D* resulted in the supernumerary spikelets (SS) phenotype at bread wheat mutant lines (*wfzp-d.1* and *wfzp-d.2*). *Dobrovolskaya et al. (2015)* also found similar mutants (Bd8202 and Bd8972) in *Brachypodium distachyon* (Bd8202 and Bd8972), showed an SS phenotype but produced

viable seeds. Two transitions were detected: G232A in Bd8972 and G163A in Bd8202. These mutations lead to amino acid substitutions (G11S and A34T in Fig. 1A) in the conserved region of the AP2/ERF domain. *Chuck (2002)* analysis of mutant alleles (*bd1-N* and *bd1-3*) revealed a transition mutation (W10Stop in Fig. 1A) within the AP2/ERF domain that introduces a premature stop codon in *bd1-N*, and deletion of the 3′ end (C-terminus) of the gene that is predicted to remove the last ten amino acids in *bd1-3*. The nine amino acid deletions (Y310_H318del in Fig. S1) in the C-terminus of the AAX83534.1 gene in *Oryza longistaminata* are considered "deleterious" (Table 3 and Fig. S1) in this research. These results indicate that the C-terminus is also very important for the biological function of the *FZP* gene. *Komatsu et al. (2003)* also believe that the C-terminus strongly indicate that this conserved region must also be necessary for proper *FZP* function in addition to the AP2/ERF domain through the study of *FZP* gene mutants.

Through previous studies on *FZP* gene mutants and our analysis of *FZP* orthologous genes, we believe that although the evolution of *FZP* genes in Gramineae species is relatively conservative, especially AP2/ERF domain, it is not ruled out that its function will change in some gramineous species. We also observed the C70 mutation in the AP2/ERF domain of the QBI22216.1 gene (*T. monococcum*) (Fig. 3 and Table 3) in the prediction results of the protein 3D structure of the 5 *FZP* orthologous genes. This mutation occurred in the seven amino acids region that specifically binds to the GCC box lead to the 3D structure of the whole protein has undergone major changes, the AP2/ERF domain cannot form an $\alpha$ helix, and the C-terminus acidic domain is far away from the AP2/ERF domain, which may directly affect the biological function of the gene.

## CONCLUSION

*FZP*, an essential gene that controls the development and the spikelet differentiation in the grass family (Poaceae), prevents the formation of axillary bud meristems and is closely associated with crop yields. It is unclear whether the *FZP* gene or its orthologs were selected during the evolutionary process of grass species, which possess diverse spike morphologies. Hence, in the present study, we identified the orthologs of the *FZP* gene in selected grass species and performed evolutionary analysis using bioinformatics methods to determine the mutations in coding DNA sequences during the evolution process of Poaceae. The phylogenetic analysis showed that the 29 Poaceae species could be classified into three groups, namely, Triticeae, *Oryza*, and C4 plants. Within the Triticeae group, the *FZP* genes originating from the same genome were classified into the same sub-group. Our analysis revealed that amino acid mutations were almost absent in the AP2/ERF domain, which indicates the highly conserved nature of the domain during the evolutionary process. We believe that our study makes a significant contribution to the literature because the results of this study may serve as a theoretical basis for understanding the functional divergence of the gene among different grass species.

## ACKNOWLEDGEMENTS

We would like to thank Editage for English language editing.

### Funding

Open Access funding was provided by Yibin Science and Technology Plan (Key) Project (No. 2018JY002). The funders had no role in study design, data collection and analysis, decision to publish, or preparation of the manuscript.

### Grant Disclosures

The following grant information was disclosed by the authors:
Open Access funding provided by Yibin Science and Technology Plan (Key) Project: 2018JY002.

### Competing Interests

The authors declare there are no competing interests.

### Author Contributions

- Jia Li and Ahmed H. El-Sappah conceived and designed the experiments, performed the experiments, analyzed the data, prepared figures and/or tables, authored or reviewed drafts of the paper, and approved the final draft.
- Litian Zhang conceived and designed the experiments, prepared figures and/or tables, and approved the final draft.
- Rania G. Elbaiomy, Bo Chen and Na Lin analyzed the data, authored or reviewed drafts of the paper, and approved the final draft.
- Lilan Chen, Zhenrong Wang and Jie Jiao performed the experiments, prepared figures and/or tables, and approved the final draft.
- Jiliang Zhu and Wanhai Zhou performed the experiments, authored or reviewed drafts of the paper, and approved the final draft.
- Salma A. Soaud and Manzar Abbas analyzed the data, prepared figures and/or tables, authored or reviewed drafts of the paper, and approved the final draft.

### Data Availability

  The raw data is available in the Supplemental Files.

### Supplemental Information

Supplemental information for this article can be found online at http://dx.doi.org/10.7717/peerj.12880#supplemental-information.

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
