# Peer review of "Evolution analysis of FRIZZY PANICLE (FZP) orthologs explored the mutations in DNA coding sequences in the grass family (Poaceae)"

_PeerJ, doi:10.7717/peerj.12880_

## Round 0.1 · original submission · Major Revisions

Dear Dr. El-Sappah,

Your article has been evaluated by the three experts in the field. Although all three were positive about your manuscript, they have raised several issues regarding your manuscript.

Please carefully address the comments of all three reviewers. Reviewer #1 raised important concerns regarding the methodology and validity of the findings that should be cautiously addressed.

Reviewer #2 pointed out missing species from the phylogenetic analysis and the need to provide details in the methods.

Lastly, Reviewer #3 raised important issues regarding grammatical mistakes particularly due to vague statements.

Also, the missing raw data should be provided in a new version of the manuscript.

Reviewer 1 ·

Basic reporting

The article is written in professional English and follows a logical structure. I have one point of criticism where changes would strongly improve the paper.

Figure 1 would be more useful if it would show an alignment between at least one candidate of each of the three subgroups with the logo shown alongside (an example would be Figure 4 in the follwoing paper: https://journals.plos.org/plosone/article?id=10.1371/journal.pone.0034231). The alignment should be extended outside of the domain and the domains clearly highlighted. It would further improve the paper if you would then mark the positive selection sites identified in section 3.4. Or even highlight the previous identified mutations which caused phenotypes in maize, rice and wheat.

The authors switch in the results over to discussion from lines 138 to 147 which is also picked up again in the discussion from line 240 to 249. In addition, the location of the genes on different chromosomes in the respective species make sense when one looks at syntenic regions between the grasses. It might be worth checking up on the synteny between grasses in the literature.

Experimental design

The methods should be further extended. Which versions of the databases were used? What was the p-value cut-off and the length identity cut-off to identify potential orthologs?
For the phylogeny it is not clear if you used the protein sequence or the CDS.

Your hypotheses from line 149 about not being able to identify genes in some of the grasses should also be moved to the discussion. I would also advise you to have another look at the data. You should be able to find out if the genomes are incomplete in the databases. Also, based on the method descriptions you seem to identify the genes based on searches against the transcriptome (which you should state in the methods), but not the genome. Incomplete transcriptomes are very common, and you might be able to find the genes with a search against the genome itself.

Validity of the findings

The last sentence of the conclusion is in no way supported by the paper and should be removed. An evolutionary analysis of one gene family is not enough to conclude anything about domestication. In general, the conclusion needs a bit more work.

Additional comments

The authors need to check their use of the word significant. Everything underpinned by a statistic analysis as for the positive selection is correct. But a mutation would only be classified as significant if an obvious phenotype could be associated with it. This is not the case for line 159, 194, 237. To estimate the effect the variance differences might have on the resulting protein, the authors could calculate PROVEAN scores to underpin their hypothesis (http://provean.jcvi.org/index.php) that those amino acid changes have an effect.

170 I assume you mean monocot species?

218 Should be “maize genome duplication”

Table 1 – why is one "NCBI" highlighted in red?

Odd changes in text size in the conclusions.

·

Basic reporting

No comments

Experimental design

No comments

Validity of the findings

No comments

Additional comments

In this manuscript, authors have used bioinformatics methods for evolutionary analysis of 35 FZP orthologs identified in 29 species of the grass family. Authors have shown that phylogenetic analysis classified these FZP orthologs into three groups (Triticeae, Oryza, and C4 plants). Also, authors have found significant difference in the acidic C-terminal regions of these three groups. Moreover, authors performed selection pressure analysis and found significant positive selection sites in species of the Triticeae and Oryza groups. This manuscript provide theoretical aspect for understanding the functional divergence of the FZP orthologs among different grass species. Overall, the manuscript is fine, and I have few comments:

1. Details about MEME is missing. For example, maximum and minimum motif length, maximum number of motifs and motif sites to be distributed in sequences.
2. Details about phylogenetic reconstruction are completely lacking for Mrbayes. What model was used to construct phylogeny?? What was the sampling frequency?? What percentage of burn-in generation was taken??
3. Line 91 & 92 Using the protein sequence of the FZP gene from rice as the seed sequence, searches were performed in the PlantGDB and other databases using what??? blastp/blastx/tblastx?? authors should mention this and what cutoff was taken to search FZP orthologs??
4. Line 130: which -> , which???
5. Line 169-170: "Using MEGA X, the optimum tree construction model T92+G was identified to generate the ML tree for the FZP gene in dicot species". I did not find any dicot species in fig2. Authors should correct it. Moreover, in method section, authors have mentioned that they have used MEGA X and Mrbayes. But there is only one phylogenetic tree in manuscript. Phylogeny present in fig2 generated by MEGA X or Mrbayes?? authors should write it in the figure2 legend. Also, authors should add another phylogenetic tree as supplementary figure.
6. I think authors should include multiple sequence alignment (MSA) as supplementary figure so that positive selection sites and other important residues that authors have mentioned could easily seen in MSA. It would be great if authors could mark those important residues in MSA with protein residues numbers.
7. For the sake of consistency, Sorghum bicolor should be referred to as S. bicolor after its first occurrence in the text. Authors should check this for all species.

·

Basic reporting

This manuscript contains mostly clear and unambiguous, professional English throughout, but there are many grammatical and contextual issues in places. Examples include,
Line 58-59: “Maize bd1 mutants are characterized by indeterminate branches in place of female spikelets, the production of indeterminate branches, and a series of lateral spikelets in tassels.“

Line 94-96: “The retrieved orthologs were identified as sequence-wise for the elimination of incomplete and redundant sequences.”

Line 236-238: “However, the rich evolutionary information of the various species has been retained through significant mutations in the acidic C-terminal region of the gene in species of the tribe Triticeae, genus Oryza, and C4 plants, which can be used for phylogenetic analyses.“

Revision for grammatical errors and more detail to slightly vague statements will bolster the quality of the manuscript.

Literature references are used in sections of the manuscript to various degrees, but are missing through as a whole, particularly with information in the results that are not from the data of this study (i.e. Line 238-240: “Previous research has indicated that foxtail millet and rice diverged approximately 50 million years ago, but the two species' genomic structures remained highly collinear.”).

The article structure, figures and tables are professional, but no raw data are provided and many other materials are missing. Only one phylogenetic tree image is used in a figure, but the underlying tree file and sequence alignment to generate tree are not provided.

The results of the study are relevant to the hypothesis presented and is self-contained.

Experimental design

The research presented in the manuscript is original, within the aims and scope of the journal, and consists of a well-defined question, relevant and meaningful to provide an understanding of the evolutionary history of FZP orthologs and the consequence on inflorescences development across grasses.
However, the study and manuscript are underdeveloped and lacking several important components.
Line 91: “Using the protein sequence of the FZP gene from rice as the seed sequence,”
• Which rice species? Two rice species included in study, Table1.
Line 91-94: “searches were performed in the PlantGDB (http://www.plantgdb.org/), Phytozome (https://phytozome.jgi.doe.gov), and NCBI (http://www.ncbi.nlm.nih.gov/) databases for orthologs contained in grass species (the corresponding CDSs were also saved).”
• Several versions of Phytzome and other databases exist, version information is missing from this study.
• It is unclear if searches were performed in only grass species or only homologs from grass species were used downstream after search. Search parameters are not provided or described.
Line 98: Dicots are mentioned, as well as in Line 128+170 and Figure 2 title, but dicots are not included in study.
Line 124-125: “we evaluated which of four codon substitution models (Model A, Model B, Model C, Model D)”
• Only Model A results reported
• Branch-Site model tests for positive selection typically involve testing different initial omega(w) to test the robustness of the analyses. The authors do not indicate if this was done in their study.
Line 155: “with mutations only occurring at positions 4 and 30.”
• It is unclear if these mutation sites are present in a single or multiple homolog sequence.
Line 170-172: “Results indicate that the grass species could be classified into groups I, II, and III. The bootstrap values of the three groups were 89%, 93%, and 68%, respectively, which were >50% and indicative of high reliability.”
• It is unclear if the groups being classified are in reference to FZP homologs or the grass species themselves.
• The bootstrap values described do not match the phylogenic tree in Figure 2
• Is the Figure 2 phylogeny based on protein or nucleotide sequence? The whole sequence or portion of alignment?
• The interpretation of bootstrap support values is inaccurate. Bootstrap values less than 95% can still provide some confidence in a branching pattern but anything below 75% is not considered reliable or “good”. (Hillis and Bull (1993) Systemic Biolgy)
Line 178-180: “Group Ⅱ was comprised entirely of species of the genus Oryza and had a bootstrap value of 93%, which indicates the extremely high reliability of the group.”
• Group II as described in Figure 2 is a paraphyletic group of Oryza and Leersia homologs, and the support value described in the manuscript for this group do not match the figure.
Line 216-218: “This indicates that the BD1B gene maybe a tandem repeat of BD1 generated during the maize genome's replication.”
• The authors have the resources to determine if the maize BD1A and BD1B are located in tandem.
• The maize homologs are not sister in the phylogenetic tree in Figure 2, suggesting a more complex evolutionary history than simple tandem duplication through replication or extensive evolution between possible tandem homologs.
Line 231: “FZP ortholog that only existed in plants”
• Were grasses he only species analyzed in this study? It would be expected that only plant homologs would be present if this is true.

The materials and methods are insufficient in information and detail to replicate the study confidently. Only one phylogenetic tree is provided, which displays data inconsistent with statements made the manuscript. This phylogenetic tree, as well as any others produced during this study, should be provide or made accessible through a public repository, such as TreeBase (https://www.treebase.org/treebase-web/home.html). Additionally, the alignments generated and used to construct the phylogenetic tree(s) should be treated similarly. Homolog searches and PAML analyses described are lacking detail in terms of the parameters used during those processes.

Validity of the findings

The finds of this study appear to be valid based on the limited data provided. The data sources used are publicly available, therefore someone could obtain the sequences used and repeat the study. However, without the parameter information for programs used and the lack of raw data corresponding sequence alignments and phylogenetic trees, it would be impossible to confidently replicate the study and results. As long as the results and finds are consistent with the underlying raw data, providing the underlying raw data will strongly bolster the quality and transparency of the manuscript. The brief conclusions drawn are supported by the data and relate to the question originally proposed.

Additional comments

This investigation into the study of FZP evolution and possible functional roles in reproductive development provides critical information as to the evolutionary role FZP has played. The results description and conclusions are basically statements of the data described in the tables. More thoughtful interpretation of the results will strengthen the impact of the study and manuscript.

---

## Round 0.2 · Minor Revisions

Dear Dr. El-Sappah,

Please follow the comments of both reviewers particularly regarding typographical errors, lack of appropriate spaces between words, which are found throughout the manuscript. Reviewer 2 also pointed out small mistakes that should be addressed before publication.

·

Basic reporting

The authors have made suggested changes and I accept the manuscript with very few minor changes. these changes are below:
line 52- Oryzasativa -> Oryza sativa??
line 60- spikeletsand -> spikelets and??
Line 257: In a specific Poaceae species category, the acidic domainofC -> domain of C (space missing). These kinds of missing spaces are a lot in the manuscript. The authors should correct it.
Line 131- Then use PyMOL v2.5.0 software(Schrodinger 2015) -> Then we did use???
Line 145: Do not start a sentence with a numeral
Line 256: The amino acids of position -> The amino acids of positions??
Line 370: Use PROVEAN online server -> We used PROVEAN online server or PROVEAN online server was used ???
Line 390-391: Use I-TASSER online server -> We did use I-TASSER or I-TASSER online server was used ??
Line 398: red area in 3D structure -> what is the red area in 3D structure. is it alpha-helix, beta-sheet, or loop region??
Line 399-401: I can not see C-terminus in 3D structure as the authors have mentioned: "The 3D structure of Arabidopsis AtERF96 gene does not have an acidic domain at the C-terminus, and gramineous plants all have an acidic domain (the orange region in the 3D structure)at C-terminus". The authors should show C-terminal in figure 3.
Line 400: the orange region in the 3D - > Authors should mention clearly what orange region is alpha-helix/beta-sheet or loop. It is good to call how these regions are called in the 3D protein structure terms.

Experimental design

no comment

Validity of the findings

no comment

·

Basic reporting

no comment

Experimental design

no comment

Validity of the findings

no comment

Additional comments

The author's have addressed and corrected most of the comments and concerns presented following the initial review of this manuscript. However, typographical errors, primarily the lack of appropriate spaces between words, are found throughout the manuscript. The Materials and Methods section has been updated and corrected based on comments from the reviewers. However, section 2.4 and 2.7 address the same program and tests of evolutionary selective pressure, and should be combined and consolidated. Lastly, I strongly encourage the author's to review the underlying assumptions of their process. The author's state that FZP "only existed in plants", but only grass (Poaceae) species were included in this study. They have not demonstrated that other plants or organisms lack FZP orthologs. Additionally, the interpretation of phylogenetic results are slightly inaccurate. As mentioned in the initial review of this manuscript, bootstrap values less than 95 can still provide some confidence in a branching pattern but values between 95 and 75 are not typically considered "indicative of high reliability", and a softer stance on values in this range should be incorporated. Bootstrap values >75 but less than 95 are "good" but not highly reliable.

---

## Round 0.3 · accepted · Accept

Dear Dr. El-Sappah,
Your manuscript has been accepted

Best regards

LINE NO: / BEFORE / AFTER / [COMMENTS]
LINE 1: / explored / explore / [.]
LINE 65: / distachyonhas / distachyon has / [.]
LINE 99: / blastp were / BLASTP analyses were / [.]
LINE 100: / Use Jalview / Use of Jalview / [.]
LINE 100: / to beautify / was used to beautify / [.]
LINE 105: / tree, used full-length / tree, using full-length / [.]
LINE 108: / Use Gblocks / Gblocks / [.]
LINE 108: / to select / was used to select / [.]
LINE 109: / sequence, MrModeltest / sequence, and MrModeltest / [.]
LINE 109: / selects the / selected the / [.]
LINE 110: / model and calculates / model to calculate / [.]
LINE 111: / reconstructs Bayesian / reconstructed a Bayesian / [.]
LINE 112: / chains are set up, / chains were set up, / [.]
LINE 122: / which of codon / which of the codon / [.]
LINE 125: / we use the custom / we used the custom / [.]
LINE 132: / We utilize the / We utilized the / [.]
LINE 134: / affects the biological / affected the biological / [.]
LINE 135: / reference NCBI nr / referenced the NCBI nr / [.]
LINE 139: / visualize according / visualize the peptide according / [.]
LINE 160: / indicate that confer / indicate sites which confer / [.]
LINE 166: / position 16 was deletion / position 16 was a deletion / [.]
LINE 201: / Dirichlet (1, 1, 1, 1)), / Dirichlet (1, 1, 1, 1)). / [period]
LINE 202: / by Bayesian / by the Bayesian / [.]
LINE 222: / protein. We used PROVEAN / protein we used the PROVEAN / [.]
LINE 225: / asterisks) detected / asterisks) were detected / [.]
LINE 225: / above -2.5, variants / above -2.5, and variants / [.]
LINE 250: / gene causes a / gene caused a / [.]
LINE 255: / highly identity FZP / highly identical FZP / [.]
LINE 286: / (Setariaitalica) / (Setaria italica) / [.]
LINE 287: / the researchers observed / the research observed for / [.]
LINE 306: / or indel has / or indels have / [.]
LINE 307: / protein. We use / protein we used / [.]
LINE 314: / Triticeae dumort / Triticeae Dumort / [as in Fig. 2]
LINE 345: / gramineous / . / [graminaceous?]